# Chitosan-Induced Physiological and Biochemical Regulations Confer Drought Tolerance in Pot Marigold (*Calendula officinalis* L.)

Gulzar Akhtar [1,*], Hafiz Nazar Faried [1], Kashif Razzaq [1], Sami Ullah [1], Fahad Masoud Wattoo [2], Muhammad Asif Shehzad [3], Yasar Sajjad [4], Muhammad Ahsan [5], Talha Javed [6,7,*], Eldessoky S. Dessoky [8], Nader R. Abdelsalam [9] and Muhammad Sohaib Chattha [10]

1   Department of Horticulture, MNS University of Agriculture, Multan 66000, Pakistan; nazar.farid@mnsuam.edu.pk (H.N.F.); kashifrazzaq@gmail.com (K.R.); sami.ullah@mnsuam.edu.pk (S.U.)
2   Department of Plant Breeding and Genetics, PMAS Arid Agriculture University, Rawalpindi 43600, Pakistan; fahad.pbg@uaar.edu.pk
3   Department of Agronomy, MNS University of Agriculture, Multan 66000, Pakistan; asif.shehzad@mnsuam.edu.pk
4   Department of Biotechnology, COMSATS University Islamabad-Abbottabad Campus, Abbottabad 22010, Pakistan; yasarsajjad@cuiatd.edu.pk
5   Department of Horticultural Sciences, The Islamia University of Bahawalpur, Bahawalpur 63100, Pakistan; ahsan.horti@iub.edu.pk
6   Department of Agronomy, University of Agriculture Faisalabad, Faisalabad 38040, Pakistan
7   College of Agriculture, Fujian Agriculture and Forestry University, Fuzhou 350002, China
8   Department of Biology, College of Science, Taif University, Taif 21944, Saudi Arabia; es.dessouky@tu.edu.sa
9   Agricultural Botany Department, Faculty of Agriculture (Saba Basha), Alexandria University, Alexandria 21531, Egypt; nader.wheat@alexu.edu.eg
10  School of Plant, Environmental, and Soil Sciences, Louisiana State University AgCenter, Baton Rouge, LA 70803, USA; sohaibchattha08@gmail.com
*   Correspondence: gulzar.akhtar@mnsuam.edu.pk (G.A.); mtahaj@fafu.edu.cn or talhajaved54321@gmail.com (T.J.)

**Abstract:** Severe water stress conditions limit growth and development of floricultural crops which affects flower quality. Hence, development of effective approaches for drought tolerance is crucial to limit recurring water deficit challenges. Foliar application of various plant growth regulators has been evaluated to improve drought tolerance in different floricultural crops; however, reports regarding the role of chitosan (Ci) on seasonal flowers like calendula are still scant. Therefore, we evaluated the role of Ci foliar application on morphological, physiological, biochemical, and anatomical parameters of calendula under water stress conditions. Different doses of Ci (0, 2.5, 5, 7.5, 10 mg L$^{-1}$) were applied through foliar application to evaluate their impact in enhancing growth and photosynthetic pigments of calendula. The optimized Ci level of 7.5 mg L$^{-1}$ was further evaluated to study mechanisms of water stress tolerance in calendula. Ci application significantly increased biomass and pigments in calendula. Ci (7.5 mg L$^{-1}$) resulted in increased photosynthetic rate (72.98%), transpiration rate (62.11%), stomatal conductance (59.54%), sub-stomatal conductance (20.62%), and water use efficiency (84.93%). Furthermore, it improved catalase, guaiacol peroxidase, and superoxide dismutase by 56.70%, 64.94%, and 32.41%, respectively. These results highlighted the significance of Ci in inducing drought tolerance in pot marigold.

**Keywords:** annual flower; floriculture; physiology; drought; chitosan

## 1. Introduction

Calendula (*Calendula officinalis* L.), commonly known as pot marigold, belongs to the Asteraceae family of Mediterranean origin [1]. It is a widely cultivated herbaceous annual or perennial (short period) with a variety of ornamental, medicinal, and cosmetic uses [2].

Pot marigold contains various secondary metabolites (flavonoids, carotenoids, steroids, terpenoids) that are a potential source of antioxidants and is used as an antibacterial, antiviral, antitumor, and anti-inflammatory [3–5]. Moreover, flower carotenoids (yellow color) are historically used as a coloring element and perfume essence. Recently, calendula has been used as an oilseed crop due to the presence of conjugated fatty acids and $\alpha$- and $\gamma$-tocopherols, used in paint and food industries [2–6]. Ornamental varieties are commonly grown for cut flower, and pot and border plants under regular irrigation and nutrient availability [7].

Irrigation management is very critical for seasonal potted plants. Limited water availability or water stress may reduce plant growth and flower yield and changes water content, chlorophyll, photosynthesis, and enzymatic activity [8,9]. Water deficit conditions also produces reactive oxygen species (ROS) like hydrogen peroxide and hydroxyl radicals that may damage proteins, lipids, and nucleic acids and ultimately affect the photosynthetic apparatus and ATP synthesis in the plant [10]. To reduce the effect of water stress, plants develop extensive roots, reduce the number of stomata, decrease tissue water potential by accumulating solvents, and increase activities of antioxidative enzymes [11]. Currently, the application of novel substances like seaweed extract, myo-insitol, trinexapac-ethyl, nanoparticles, aluminosilicate (kaolin), and chitosan (Ci) is extensively used to mitigate effects of water deficit conditions in flowering plants [12].

Ci is an effective natural marine bio-stimulant polysaccharide manufactured by alkaline deacetylation of chitin and is available in liquid or powder form. Ci is effective against biotic and abiotic stresses in plants. It is known to increase plant growth regulators such as indole acetic acid, gibberellin, and abscisic acid that protect plants from oxidative stress and increases crop yield [13–15] under water stress conditions. Previous studies by Tourian et al. [16] and El-Serafy [17] indicated that Ci increased shoot and root growth, photosynthetic pigments, and plant quality under water stress conditions by regulating phenolic and enzymatic activities. The positive role of Ci in increasing water stress tolerance is well reported in ornamental plants like sage (*Salvia officinalis*) [18], freesia (*Freesia odorata*) [19], basil (*Ocimum ciliatum* and *Ocimum basilicum*) [20], cordyline (*Cordyline fruticosa*) [17], Chrysanthemum (*Chrysanthemum morifolium*) [21], and sunflower (*Helianthus annuus*) [10].

Hence, an effective bio-stimulative and metabolic profile of Ci and its role in water stress tolerance enthused us to evaluate the foliar application of different Ci levels on calendula plant under water deficit conditions. Ci application has been reported in different horticultural crops, but its role in mitigating water stress in calendula plants has not been studied yet. The present research work is focused on the physiological, biochemical, and anatomical alterations, vital for increasing tolerance against water stress in potted calendula plants.

## 2. Materials and Methods

### 2.1. Planting Material and Conditions

The pot experiment was conducted at the research area Muhammad Nawaz Shareef University of Agriculture Multan (31°30′ N, 73°10′ E, elevation 213 m). Healthy seeds of calendula cultivar 'Orange King' were obtained from the Pak Green Seed Store in Lahore, Pakistan. Seeds were sown (6 cm depth) in the center of earthen pots (24 cm width and 30 cm length) filled with air dried, sieved (2 mm mesh), and thoroughly mixed potting media (3 kg pot) containing 70% silt and 30% sludge. According to soil analysis, the potting media contains pH 8.0, organic matter 0.39%, electrical conductivity of saturated extract (ECe) 10.91 mS cm$^{-1}$, saturation percentage 18, total available N 0.021%, available $p$ 5.20 mg kg$^{-1}$, and available K 110 mg kg$^{-1}$. The pots were placed under natural conditions (Supplementary Table S1) with rain out shelter protection during the growth period (December 2019–March 2020).

*Experiment I: Optimization of chitosan dose*

For dose optimization of Ci, two seeds were sown in each pot and were thinned to one healthy seedling. At 6 leaf stage, different Ci (0, 2.5, 5, 7.5, 10 mg L$^{-1}$) levels were applied twice at a seven days interval. Ci (Bio Basic Inc., Markham, ON, Canada) was dissolved in 1% acetic acid solution and distilled water was used for making different dilutions, then 0.1% Tween-20 was added as surfactant. Control plants were foliar sprayed with distilled water. After 15 days of the 2nd Ci application, data of different growth and photosynthetic pigments were recorded. To analyze data, completely randomized design (CRD) in three replicates was used and means were compared using least significance difference (LSD).

*Experiment II: Foliar application of optimized chitosan dose on calendula*

The impact of optimized Ci level (7.5 mg L$^{-1}$) (Supplementary Tables S2 and S3) on physio-biochemical and anatomical attributes of calendula was analyzed under drought conditions. Plants were grown under normal conditions till 6 leaf stage and pots were divided in 4 groups as W for control, or normal conditions (100% FC; Field capacity and no Ci), W + Ci for 100% FC and Ci, D for drought stress (60% FC and no Ci) and D + Ci for drought stress (60% FC) and Ci. The moisture content of wet soil was determined by gravimetric method and their levels (100% and 60% FC) were maintained by daily weighing of pots to determine water loss and supplementing of required quantity of water to maintain a constant FC [10]. Drought stress (60% FC) was applied to plants at the six-leaf stage, then after a week two, foliar applications of optimized Ci level (7.5 mg L$^{-1}$) were applied at a one-week interval using a hand sprayer of 1 L capacity between 7 to 10 a.m. Eight weeks after initial treatment, plants (1 plant in each pot) were harvested for subsequent measurements. This experiment had four treatments (two factors, i.e., Ci and water regime as $2 \times 2$ factorial) and arranged under CRD design in three replicates.

## 2.2. Measurement of Growth Characteristics

The number of leaves (NOL) per plant was counted manually and leaf area (LA) was determined using a leaf area meter (Model CI-202, CID Inc., Camas, WA, USA) [22]. At the end of the experiment, plants were harvested and washed to measure shoot length (SL) and root length (RL) using a meter rod. Shoot and root fresh weights (SFW and RFW) were measured immediately after separating into shoot and root that were further oven dried (Memmert GmbH + Co.KG, Model 30–750, Germany) at 65 °C for 72 h to determine shoot and root dry weights (SDW and RDW).

## 2.3. Estimation of Leaf Chlorophyll Pigments and Color Intensity Value

The photosynthetic pigments chlorophyll a (Chl *a*), chlorophyll b (Chl *b*), and carotenoids (Car) were estimated according to [23]. In brief, 0.5 g leaf sample was thawed, grinded in 80% acetone and placed overnight at 4 °C. Then, a crude sample was centrifuged at 1500 rpm for 5 min and the supernatant was used for absorbance reading using a spectrophotometer at 645, 663, and 480 nm, denoted by OD645, OD663, and OD480, respectively. The V and W represented volume supernatant and weight of leaf sample, respectively. These absorbance readings were used to calculate Chl *a*, *b* and Car, as follows:

$$\text{Chl } a \text{ (mg g}^{-1} \text{ FW)} = [12.7 \, (\text{OD663}) - 2.69 \, (\text{OD645})] \times (\text{V}/1000) \times \text{W}$$

$$\text{Chl } b \text{ (mg g}^{-1} \text{ FW)} = [22.9 \, (\text{OD645}) - 4.68 \, (\text{OD663})] \times (\text{V}/1000) \times \text{W}$$

$$\text{Car (μg g}^{-1} \text{ FW)} = \text{A}^{\text{car}}/\text{E max100}$$

$$(\text{A}^{\text{car}} = [(\text{OD480}) + 0.114 \, (\text{OD663}) - 0.638 \, (\text{OD645})] \text{ and E max100 cm} = 2500)$$

The color of the intact outer leaf surface was quantified using chromameter (CR-400 Konica Minolta Bench-top, Tokyo, Japan) by calculating three color coordinates: L* represents brightness/lightness (higher value denotes brightness), a* represents redness/greenness (negative denotes green and positive red color), and b* represents yellowness/blueness (negative value showed blue and positive to yellow color) [24].

### 2.4. Estimation of Leaf Water Status and Membrane Stability Index

For estimation of water contents (WC) and relative turgidity (RT), turgid young leaves were immediately weighed to get fresh weight (FW). Afterwards same leaves were soaked in distilled water (4 °C for 24 h) to determine turgid weight (TW) and then oven-dried (65 °C, 72 h) to estimate dry weight (DW). The WC and RT were calculated using formulas suggested by Redondo-Gomez et al. [25] and Clausen and Kozlowski [26]

$$WC\ (\%) = (FW - DW/FW) \times 100$$

$$RT\ (\%) = [(FW - DW)/(TW - DW)] \times 100$$

The excised leaf water retention (ELWR) and excised leaf water loss (ELWL) were measured from young turgid leaves that were immediately weighed to record fresh weight (FW). Then, leaves were placed at room temperature (25 °C) for 6 h to record weight loss (WL) to calculate ELWR. Leaves were also incubated for 6 h (20 °C, 50% humidity) to determine incubation weight (IW) to calculate ELWL. Dry weight (DW) of leaves was calculated after oven drying at 65 °C for 72 h and the following formulas recommended by Lonbani and Arzani [27] and Clarke and McCaig [28] were used to measure ELWR and ELWL:

$$ELWR\ (\%) = [1 - (FW - WL)/FW] \times 100$$

$$ELWL\ (\%) = (FW - IW)/(FW - DW) \times 100$$

To measure membrane stability (MSI), two leaf samples (0.2 g) were taken before termination of the experiment and rinsed in 20 mL distilled water in two 50 mL volumetric flasks. Then, both flasks were placed in a hot water bath to take two electrical conductivity readings, first at 40 °C after 30 min (C1) and second at 100 °C after 15 min (C2), and MSI was calculated using the formula by Sairam et al. [29].

$$MSI = [1 - (C1/C2)] \times 100$$

### 2.5. Determination of Gas Exchange Parameters

Fully expanded mature leaves were used to record net photosynthetic rate ($A$), transpiration rate $E$, stomatal conductance ($gs$), sub-stomatal conductance ($Ci$), and water use efficiency (WUE) using a CIRAS-3 portable open-flow gas exchange system (PP Systems, Amesbury, MA, USA) between 11 a.m. to 1 p.m.

### 2.6. Determination of Antioxidant Enzymes Activity

For determination of catalase (CAT), guaiacol peroxidase (GPX) superoxide dismutase (SOD) activities, fresh leaf samples (0.5 g) were homogenized using mortar and pestle with phosphate buffer (pH 7.0). Then, samples were centrifuged at 1500 rpm for 15 min and supernatant was separated to quantify the activities of CAT, GPX, and SOD.

CAT activity was assessed using the procedure of Chance and Maehly [30] in which supernatant (0.1 mL) along with phosphate buffer (pH 7.0) and $H_2O_2$ (5.9 mM) were mixed and absorbance was read at 240 nm using the spectrometer (Hitachi-220, Japan).

The procedure reported by Urbanek et al. [31] was used to determine GPX activity, in which a 2 mL reaction sample (50 mM phosphate buffer, 20 mM guaiacol, 40 mM $H_2O_2$ and 0.1 mL supernatant) was used for measuring absorbance at 470 nm.

The enzymatic activity of SOD was measured using the procedure of Van Rossun et al. [32] in which 50 µL supernatant with 50 mM, potassium phosphate buffer (7.8 pH), 2 µM riboflavin, 100 µM EDTA, and 75 µM p-nitroblue tetrazolium chloride was assessed under white fluorescent light (30 W, 10 min) and absorbance measured at 560 nm.



### 2.7. Measurement of Stomatal Density and Area

Stomatal density and area were measured from the abaxial side of the young leaves according to the procedure reported by Omidbaigi et al. [33]. Before harvesting of plants, expanded young leaves were selected and nail varnish applied to their abaxial surface and allowed to dry for 20 min. Then, the dried nail varnish film along with an imprint of epidermis layer were removed and placed on a glass slide that adjusted on a light microscope (XSZ-107BN, USA) fitted with an ocular micrometer to measure stomatal density and area.

### 2.8. Statistical Analysis

Analysis of variance (ANOVA) was performed using STATISTIX (Version 8.1). Furthermore, LSD test was used to determine significance among treatment means.

## 3. Results

*Experiment I: Dose optimization for chitosan*

All growth parameters of calendula were significantly improved in response to foliar application of different Ci levels (Supplementary Table S2). Ci at 7.5 mg $L^{-1}$ increased the number of leaves by 15.78% compared to control (No Ci), whereas a higher level of Ci (10 mg $L^{-1}$) decreased the number of leaves by 6.28% in contrast to control. The shoot and root lengths of plants were enhanced by 15.78% and 45.44%, respectively, in response to a 7.5 mg $L^{-1}$ dose of Ci with respect to control. Ci at 10 mg $L^{-1}$ reduced shoot length (37.50%) and root length (46.25%) compared to No Ci (control). Foliar application of Ci at 7.5 mg $L^{-1}$ maximum increased the shoot and root fresh weights by 60.51% and 9.49% which further decreased by 10.10% and 59.88% at 10 mg $L^{-1}$ Ci, respectively, compared to control. The highest shoot and root dry weights by 61.41% and 22.38% were noted in Ci (7.5 mg $L^{-1}$) treated plants with respect to control. Plants exhibited a marked decline in shoot and root dry weights by 12.20% and 63.46%, respectively, at 10 mg $L^{-1}$ Ci over the control (No Ci).

Ci foliar application significantly increased the photosynthetic pigments in calendula leaves. Chl *a*, Chl *b*, and Car were considerably improved by 46.26%, 47.56%, and 30.55%, respectively, with Ci at 7.5 mg $L^{-1}$ followed by Chl *a* (36.84%), Chl *b* (40.28%), and Car (20.63%) at 5 mg $L^{-1}$ in contrast to control (No Ci) (Supplementary Table S3).

*Experiment II: Foliar application of optimized chitosan dose on calendula*

### 3.1. Biomass Attributes

Water stress of 60% FC caused a significant decline in the number of leaves and leaf area of calendula plants by 20% and 12.53%, respectively. Ci foliar application considerably increased the number of leaves and leaf area by 34% and 25%, respectively, under normal conditions. A substantial improvement in the number of leaves (30% for 7.5 mg $L^{-1}$ Ci) and leaf area (16.52% for 7.5 mg $L^{-1}$ Ci) was recorded compared to control (0 mg $L^{-1}$ Ci) under water deficit conditions. A significant decline in shoot and root lengths by 9.50% and 26.09% was observed under water stress (60%); while a considerable increase by 28.30% and 37.04%, respectively, was noticed for 7.5 mg $L^{-1}$ Ci. Ci application significantly improved the shoot and root lengths by 41% and 41%, respectively, under normal conditions. Similarly, Ci foliar application substantially improved in shoot and root dry weights, 54.46% and 48.98%, respectively, under water stress compared to the control (0 mg $L^{-1}$ Ci). Supplemented Ci considerably increased the flower dry weight by 32.63% for 7.5 mg $L^{-1}$ to water stress of 60% FC over control (No Ci) (Table 1). Foliar Ci application considerably enhanced the shoot, root, and flower dry weights by 72%, 67%, and 53%, respectively.

**Table 1.** Number of leaves (NOL), leaf area (LA), shoot length (SL), root length (RL), shoot dry weight (SDW), root dry weight (RDW), and flower dry weight (FDW) of *Calendula officinalis* applied with foliar application of chitosan (7.5 mg L$^{-1}$) at 100% FC (W), 60% FC (D).

| Treatments | NOL | LA (cm$^2$) | SL (cm) | RL (cm) | SDW (g) | RDW(g) | FDW |
|---|---|---|---|---|---|---|---|
| W | 35 $^c$ ± 1.76 | 3910 $^b$ ± 30.41 | 14.00 $^c$ ± 0.34 | 23 $^c$ ± 1.18 | 12.98 $^c$ ± 0.19 | 9.00 $^c$ ± 0.39 | 0.84 $^b$ ±0.04 |
| W + Ci | 53 $^a$ ± 0.34 | 5195 $^a$ ± 58.87 | 23.67 $^a$ ± 0.34 | 39 $^a$ ± 0.59 | 46.88 $^a$ ± 0.40 | 27.33 $^a$ ± 0.39 | 1.77 $^a$ ± 0.07 |
| D | 28 $^d$ ± 1.48 | 3420 $^c$ ± 183.42 | 12.67 $^c$ ± 0.90 | 17 $^d$ ± 1.18 | 12.77 $^c$ ± 0.69 | 5.78 $^d$ ± 0.41 | 0.64 $^c$ ± 0.02 |
| D + Ci | 40 $^b$ ± 0.90 | 4097 $^b$ ± 46.78 | 17.67 $^b$ ± 0.90 | 27 $^b$ ± 0.90 | 29.33 $^b$ ± 0.52 | 11.33 $^b$ ± 0.71 | 0.95 $^b$ ± 0.04 |
| *p*-value | | | | | | | |
| D | <0.0001 | <0.0001 | 0.0004 | <0.0001 | <0.0001 | <0.0001 | <0.0001 |
| Ci | <0.0001 | <0.0001 | <0.0001 | <0.0001 | <0.0001 | <0.0001 | <0.0001 |
| D × Ci | 0.0755 | 0.0149 | 0.0117 | 0.0116 | <0.0001 | <0.0001 | 0.0001 |
| CV | 5.45 | 4.10 | 6.76 | 6.37 | 3.25 | 6.27 | 7.42 |

Values are mean ± SE and letters represent significant differences at *p* < 0.05 according to LSD test (each mean represents 3 replicates). CV, Coefficient of variation.

### 3.2. Chlorophyll and Leaf Color Parameters

Photosynthetic pigments (Chl *a*, Chl *b*, and Car) significantly reduced 54.39%, 21.26%, and 71.05%, respectively, in the leaves of calendula subjected to water stress of 60% FC with respect to normal plants (100 FC). Ci foliar application considerably improved the Chl *a*, Chl *b*, and Car by 29%, 32%, and 32%, respectively, under normal conditions. Foliar Ci application (7.5 mg L$^{-1}$) markedly increased the Chl *a*, Chl *b*, and Car by 56.67%, 40.48%, and 71.79%, respectively, under water stress (60% FC) (Table 2). Water stress exposure caused significant reduction in leaf color, L*, a*, and b* by 23.58%, 35.18%, and 4.52% in contrast to control (100% FC). The improved L* (27.93%), a* (37.93%), and b* (6.14) was recorded under water stress of 60% FC with foliar application of Ci at 7.5 mg L$^{-1}$ compared to control (no Ci) (Table 2).

**Table 2.** Chlorophyll (Chl *a*, Chl *b*), carotenoid (Car), L* (Brightness/Lightness), a* (Redness/Greenness), and b* (Yellowness/Blueness) content of *Calendula officinalis* treated with foliar application of chitosan (7.5 mg L$^{-1}$) at 100% FC (W), 60% FC (D).

| Treatments | Chl *a* (mg g$^{-1}$) | Chl *b* (mg g$^{-1}$) | Car (mg g$^{-1}$) | L* | a* | b* |
|---|---|---|---|---|---|---|
| W | 0.57 $^c$ ± 0.03 | 1.27 $^c$ ± 0.04 | 0.38 $^c$ ± 0.04 | 34.78 $^c$ ± 0.42 | 7.39 $^c$ ± 0.15 | 10.40 $^c$ ± 0.03 |
| W + Ci | 0.8 $^a$ ± 0.07 | 1.87 $^a$ ± 0.05 | 0.56 $^a$ ± 0.07 | 41.78 $^a$ ± 1.02 | 9.62 $^a$ ± 0.02 | 10.85 $^a$ ± 0.02 |
| D | 0.26 $^d$ ± 0.08 | 1 $^d$ ± 0.06 | 0.11 $^d$ ± 0.05 | 26.65 $^d$ ± 0.96 | 4.79 $^d$ ± 0.06 | 9.93 $^d$ ± 0.06 |
| D + Ci | 0.6 $^b$ ± 0.06 | 1.68 $^b$ ± 0.01 | 0.39 $^b$ ± 0.04 | 36.98 $^b$ ± 0.19 | 7.72 $^b$ ± 0.03 | 10.58 $^b$ ± 0.04 |
| *p*-value | | | | | | |
| D | <0.0001 | <0.0001 | <0.0001 | <0.0001 | <0.0001 | <0.0001 |
| Ci | <0.0001 | <0.0001 | <0.0001 | <0.0001 | <0.0001 | <0.0001 |
| D × Ci | 0.0004 | 0.0052 | 0.0006 | 0.0243 | 0.0046 | 0.0120 |
| CV | 2.80 | 1.20 | 4.35 | 2.75 | 1.87 | 0.49 |

Values are mean ± SE and letters represent significant differences at *p* < 0.05 according to LSD test (each mean represents 3 replicates). CV, Coefficient of variation.

### 3.3. Water Status and Membrane Stability Index

Water stress at 60% FC significantly reduced leaf WC (43.65%) and RT (66.65%) of calendula compared to control (no water stress) (Figure 1A,B). Ci application considerably improved WC (44.34%) and RT (47.04%) with respect to water stress control (no Ci application). ELWR was also markedly decreased to 77.01% whereas ELWL increased (75.88%), respectively, in water stress (60% FC) plant with respect to control (100% FC) (Figure 1C,D). Ci foliar application improved ELWR (66.19%) and reduced ELWL (41.02%) under water stress compared to control. Water stress also decreased (*p* < 0.05) MSI (70.57%) compared to normal plants (100% FC); however, Ci treatment remarkably increased MSI (71.03%) under water stress of 60% FC compared to control (no Ci) (Figure 1E). Foliar Ci application

significantly increased the WC, RT, ELWR, and MSI by 17.87%, 32.43%, 34.39%, and 51.87%, respectively, under normal condition.

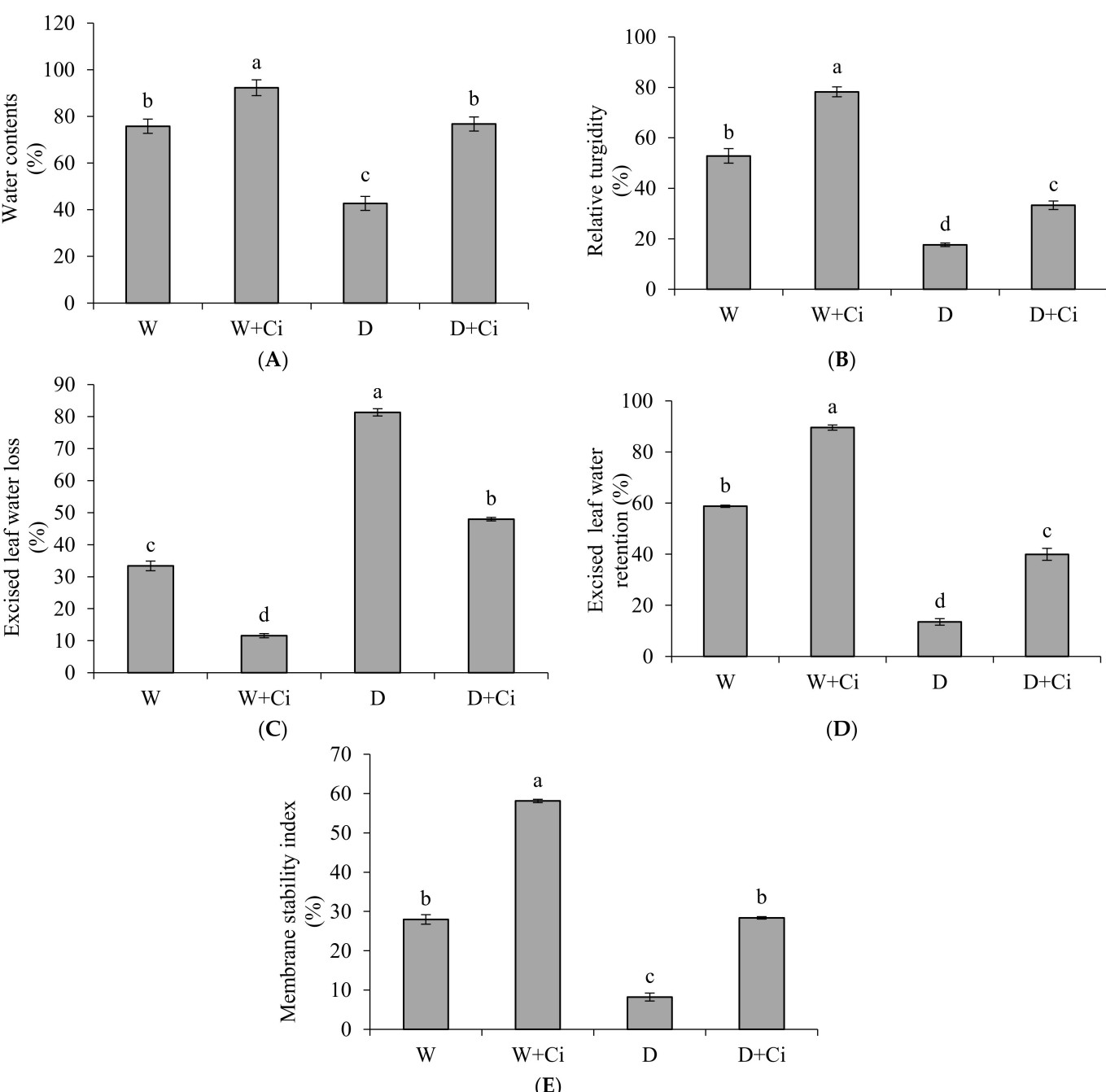

**Figure 1.** (**A**) Water contents; (**B**) Relative turgidity; (**C**) Excised leaf water loss; (**D**) Excised leaf water retention; and (**E**) Membrane stability index contents of *Calendula officinalis* affected by foliar application of chitosan (7.5 mg L$^{-1}$) weekly at 100% and 60% FC. Letters above the bars represent significant differences at $p < 0.05$ according to LSD test. W, normal conditions (100% FC); W + Ci, chitosan under normal conditions; D, drought stress without chitosan (60% FC); D + Ci, drought stress with chitosan (each mean consists of 3 replicates).

*3.4. Gas Exchange Parameters*

Water stress considerably decreased *A* and *E* by 74.75% and 50.42%, respectively, with respect to normal plants (100 mg L$^{-1}$). The *A* and *E* were significantly improved by 72.97% and 62.10% upon application of Ci at 7 mg L$^{-1}$ (Figure 2A,B) under drought stress. Ci

application also considerably improved *A* and *E* by 47.05% and 36.73%, respectively, under normal condition. The recorded values of *gs*, *Ci* and WUE showed marked reduction by 48.91%, 15.70%, and 92.25%, respectively, at 60% FC over the normal plants (Control, 100% FC); whereas increased values 59.54%, 20.54%, and 84.72% of *gs*, *Ci*, and WUE, respectively, were noted in response to foliar application Ci (7 mg $L^{-1}$) under water stress of 60% FC compared to control (No Ci) (Figure 2C–E). Significantly increased *gs*, *Ci*, and WUE by 44.99%, 13.56%, and 28.28% were also recorded in plants supplemented with Ci under normal conditions.

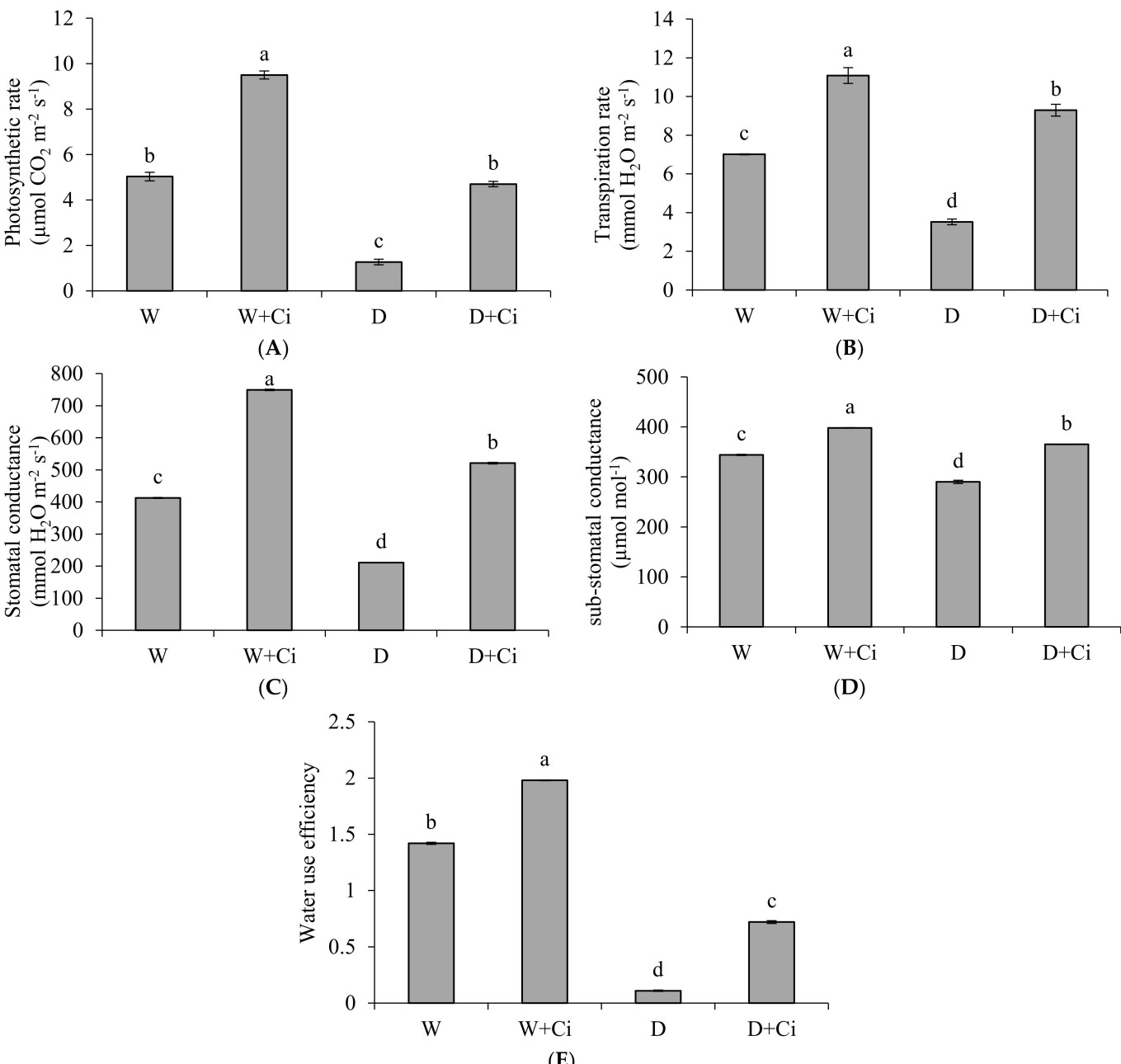

**Figure 2.** (**A**) Photosynthetic rate; (**B**) transpiration rate; (**C**) stomatal conductance; (**D**) sub-stomatal conductance; and (**E**) water use efficiency of *Calendula officinalis* after foliar application of chitosan (7.5 mg $L^{-1}$) weekly at 100% and 60% FC. Letters above the bars represent significant differences at $p < 0.05$ according to LSD test. W, normal conditions (100% FC); W + Ci, chitosan under normal conditions; D, drought stress without chitosan (60% FC); D + Ci, drought stress with chitosan (each mean consists of 3 replicates).

### 3.5. Antioxidant Enzymes

Exposure to water stress of 60% FC significantly ($p < 0.05$) increased CAT (54.76%), GPX (61.76%) and SOD (21.03%) in relation to control (100% FC) (Figure 3). Significantly improved CAT value by 56.70% was recoded in calendula plant supplemented with Ci (7 mg L$^{-1}$) and grown under water stress of 60% FC (Figure 3A). Application of foliar Ci at (7 mg L$^{-1}$) to water stress plants (60%FC) considerably enhanced GPX by 64.94% compared with control (No Ci) (Figure 3B). Water stress of 60% FC significantly increased SOD level by 32.41% by Ci 7 mg L$^{-1}$ with respect to control (0 mg L$^{-1}$ Ci) (Figure 3C). Ci application also considerably improved CAT, GPX, and SOD by 64.15%, 75.47%, and 32.15%, respectively, under normal condition.

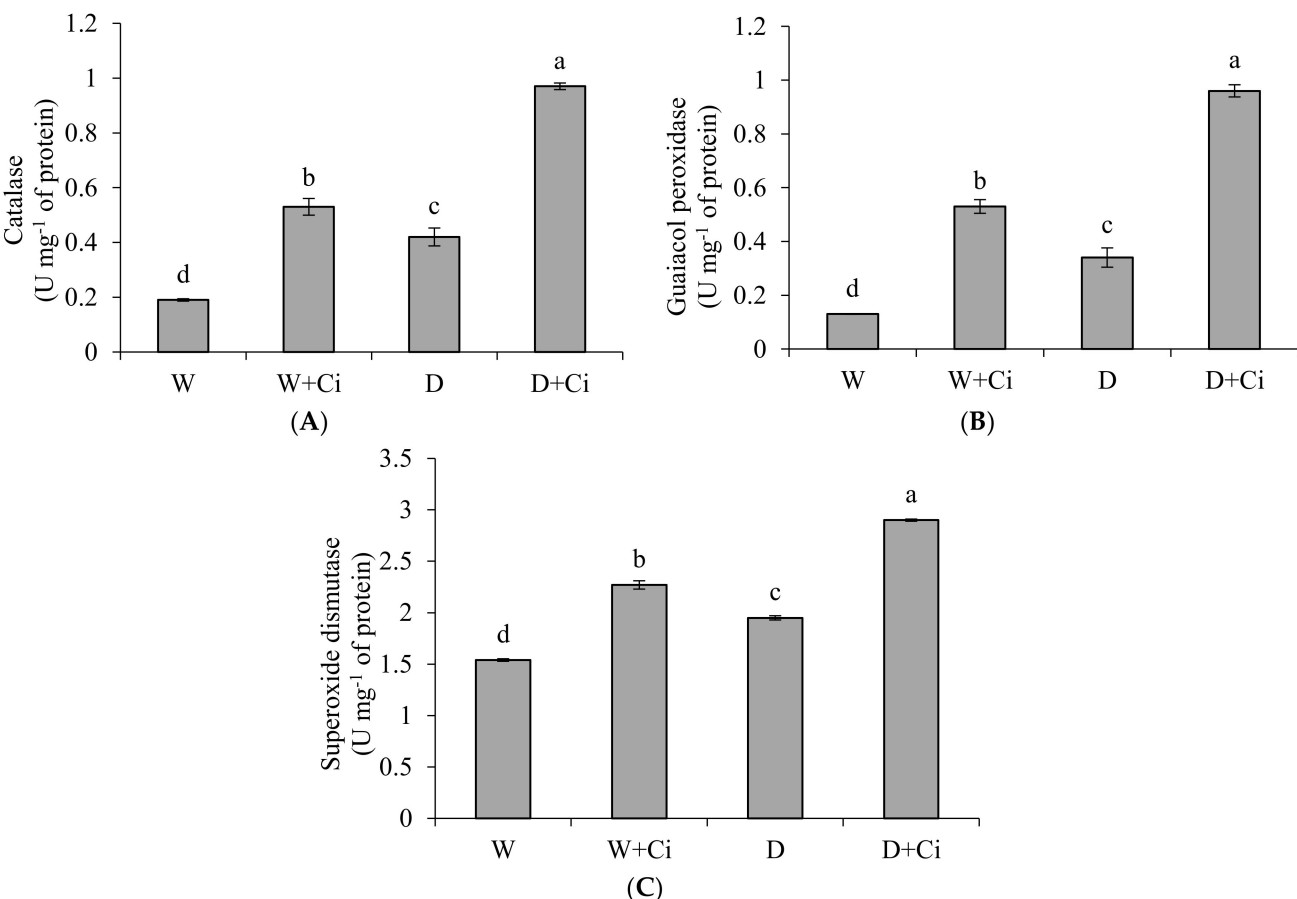

**Figure 3.** (**A**) Catalase; (**B**) guaiacol peroxidase and (**C**) superoxide dismutase activity of *Calendula officinalis* after foliar application of chitosan (7.5 mg L$^{-1}$) weekly at 100% and 60% FC. Letters above the bars represent significant differences at $p < 0.05$ according to LSD test. W, normal conditions (100% FC); W + Ci, chitosan under normal conditions; D, drought stress without chitosan (60% FC); D + Ci, drought stress with chitosan (each mean consists of 3 replicates).

### 3.6. Anatomical Parameters

Adaxial leaf stomatal anatomy of calendula was significantly ($p < 0.05$) influenced by water stress and Ci application (Figure 4). Stomatal density was considerably reduced by 78.13% under water stress of 60% FC with respect to well-watered (100% FC) plants (Figure 4A). However, application of Ci improved stomatal density by 61.11% in comparison to water stress plant (60% FC with no Ci) and 36% under normal conditions (100% FC with no Ci). Exposure to water stress of 60% FC significantly reduced stomatal area (89.72%) with respect to normal plants (100% FC). Improved stomatal area by 59.50% was recorded in plant supplemented with Ci (7 mg L$^{-1}$) under water stress of 60% FC with respect to

control (No Ci) (Figure 4B). Ci application also considerably improved stomatal area by 23.18% under normal condition compared to control (100% FC).

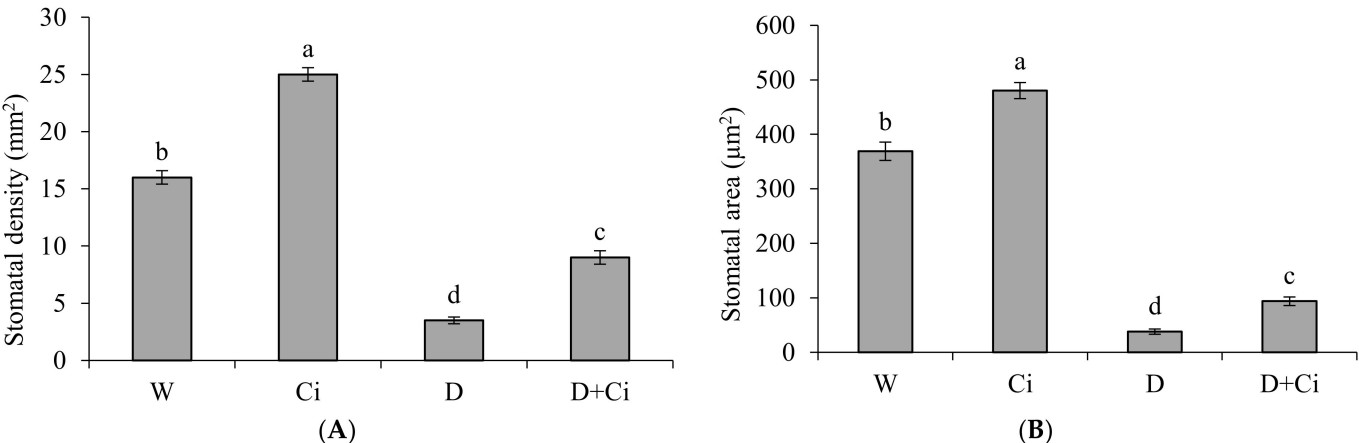

**Figure 4.** (**A**) Stomatal density and (**B**) stomatal area of *Calendula officinalis* after foliar application of chitosan (7.5 mg L$^{-1}$) weekly at 100% and 60% FC. Letters above the bars represent significant differences at $p < 0.05$ according to LSD test. W, normal conditions (100% FC); W + Ci, chitosan under normal conditions; D, drought stress without chitosan (60% FC); D + Ci, drought stress with chitosan (each mean consists of 3 replicates).

*3.7. Pearson Correlation*

The flower attributes of calendula have a significant positive correlation with growth morphological and physiological parameters, whereas nonsignificant correction with the leaf antioxidative activity of calendula (Table 3).

**Table 3.** Pearson correlation among morphological, physiological, and biochemical traits of calendula following foliar chitosan application under water stress conditions.

| | LA | SDW | RDW | SL | RL | WC | RT | SD | FDW | Chl *a* | Chl *b* | Car | SOD | GPX | CAT |
|---|---|---|---|---|---|---|---|---|---|---|---|---|---|---|---|
| LA | | | | | | | | | | | | | | | |
| SDW | 0.928 ** | | | | | | | | | | | | | | |
| RDW | 0.956 ** | 0.943 ** | | | | | | | | | | | | | |
| SL | 0.916 ** | 0.960 ** | 0.959 ** | | | | | | | | | | | | |
| RL | 0.953 ** | 0.938 ** | 0.956 ** | 0.954 ** | | | | | | | | | | | |
| WC | 0.840 ** | 0.746 ** | 0.770 ** | 0.805 ** | 0.862 ** | | | | | | | | | | |
| RT | 0.882 ** | 0.713 ** | 0.871 ** | 0.804 ** | 0.879 ** | 0.851 ** | | | | | | | | | |
| SD | 0.890 ** | 0.717 ** | 0.876 ** | 0.792 ** | 0.862 ** | 0.863 ** | 0.986 ** | | | | | | | | |
| FDW | 0.962 ** | 0.932 ** | 0.984 ** | 0.957 ** | 0.960 ** | 0.776 ** | 0.882 ** | 0.867 ** | | | | | | | |
| Chl *a* | 0.915 ** | 0.828 ** | 0.853 ** | 0.869 ** | 0.925 ** | 0.961 ** | 0.893 ** | 0.893 ** | 0.865 ** | | | | | | |
| Chl *b* | 0.886 ** | 0.929 ** | 0.845 ** | 0.917 ** | 0.923 ** | 0.877 ** | 0.718 ** | 0.715 ** | 0.849 ** | 0.928 ** | | | | | |
| Car | 0.904 ** | 0.812 ** | 0.842 ** | 0.856 ** | 0.913 ** | 0.975 ** | 0.896 ** | 0.899 ** | 0.848 ** | 0.993 ** | 0.916 ** | | | | |
| SOD | 0.280 ns | 0.551 ns | 0.254 ns | 0.439 ns | 0.351 ns | 0.254 ns | −0.095 ns | −0.103 ns | 0.252 ns | 0.295 ns | 0.613 * | 0.268 ns | | | |
| GPX | 0.277 ns | 0.523 ns | 0.220 ns | 0.406 ns | 0.319 ns | 0.268 ns | −0.104 ns | −0.106 ns | 0.223 ns | 0.303 ns | 0.606 * | 0.276 ns | 0.991 ** | | |
| CAT | 0.166 ns | 0.455 ns | 0.148 ns | 0.339 ns | 0.252 ns | 0.194 ns | −0.193 ns | −0.193 ns | 0.133 ns | 0.209 ns | 0.535 ns | 0.193 ns | 0.986 ** | 0.976 ** | |

Note: LA, leaf area; SDW, shoot dry weight; RDW, root dry weight; SL, shoot length; RL, root length; WC, water contents; RT, relative turgidity; SD, stomatal density; FDW, flower dry weight; Chl *a*, chlorophyll a; Chl *b*, chlorophyll b; Car, carotenoid; SOD, superoxide dismutase; GPX, guaiacol peroxidase; CAT, catalase. * $p < 0.05$, ** $p < 0.01$; ns, non-significant.

## 4. Discussion

Water stress conditions generally reduce plant biomass through decreased leaf water contents, chlorophyll concentrations, and enzyme activities [10]. In experiment I, the effect of different Ci foliar applications on biomass and chlorophyll concentrations of calendula seedlings are interpreted. In experiment II, the physio-biochemical and anatomical importance of Ci in water stress tolerance is discussed. Foliar Ci application significantly improved NOL, LA, SL, RL, SFW, RFW, SDW, RDW, Chl *a*, Chl *b*, and carotenoids. Due to increased biomass and chlorophyll contents with Ci application, we conclude that Ci supply improves photosynthesis and translocation of its products in the plants. Similar results were observed by Shehzad et al. [10] in sunflower. This positive role of Ci is dose-dependent because highly significant variations were observed among different levels (Supplementary Tables S2 and S3). Higher Ci level (10 mg $L^{-1}$) adversely affected growth of calendula seedlings, similarly to previous reports in *Arabidopsis* and cordyline [17–34] and this may be due to modifications in auxin synthesis and cell division through modifying homeodomain transcription factor WOX5 [34]. El-Serafy [17] and Tantasawat et al. [35] also reported reduced plant biomass in cordyline and Dendrobium at higher levels of Ci.

The application of sufficient levels of plant growth regulators (PGRs) improves the growth and production of ornamental plants [36,37]. However, the availability of PGRs in plants is highly important, and Ci particularly improves phytohormones (GA, auxins) that enhance root growth and water availability to plants under water stress conditions [38]. The results of the present study showed that water deficit conditions considerably reduced shoot and root growth. This reduction, along with plant architecture, may be due to decreased photosynthesis by limiting water contents, gas exchange, enzyme activity, and stomatal closure [10–39]. Previously, Grant et al. [8] also observed that ornamental potted plants are very sensitive to water deficit conditions. Similar to the findings of Vosoughi et al. [18] and El-Serafy [17] in sage and cordyline, our results also indicated that foliar Ci application significantly enhanced plant biomass and photosynthetic pigments (Tables 1 and 2). Ci foliar application is effectively absorbed in plant leaves and provides extra amino acid metabolic activities that enhance chlorophyll synthesis and growth [40]. Previous studies by Nahar et al. [41], Salachna and Zawadzinska [19], and Elansary et al. [21] also explained enhanced water and nutrient uptake to improve growth in Ci treated orchids, freesia, and chrysanthemum plants, respectively.

Maintenance of leaf chlorophyll content, color, WC, and petal MSI are strong indicators of drought tolerance. According to present results, water deficit conditions considerably decreased chlorophyll and carotenoid concentrations along with leaf color might be due to the degradation of pigments by excessive reactive oxygen species (ROS) production [42]. The present results are similar to those of Elansary et al. [21] in chrysanthemum and of Oraee and Tehranifar [43] in pansy which provided more evidence that water stress conditions influence photosynthesis mechanism and leaf color by oxidation of leaf pigments. Exogenous Ci application improved chlorophyll and carotenoid concentrations by increasing enzymatic activity (CAT, GPX, SOD) that helps to limit lipid peroxidation under water stress conditions (Figure 4). Similarly, Salachna and Zawadzinska [19] explained increased chlorophyll contents in potted freesia by Ci application. The increase of photosynthetic pigment and growth attributes in the present findings showed that pigment synthesis significantly enhanced the growth of water stress calendula plants. A similar correlation of higher chlorophyll and increased growth of sunflowers plant was also recorded by Shehzad et al. [10]. According to the present study, water stress conditions reduced WC and MSI might be due to osmotic imbalance and reduced photosynthetic activity [44,45]. WC is often used to determine the degree of water stress because reduced RWC causes loss of cellular turgidity and photosynthesis [46,47]. The membrane stability index is wildly used to access the effects of water stress because water shortage disrupts cell membrane and internal composition [48]. Foliar Ci application improved WC and MSI indicating that Ci treatment maintained water balance in cells by stomatal adjustment and improved water uptake from roots [49,50].

The reduced leaf gas parameters (*A*, *E*, *gs*, *Ci*, WUE) of calendula under drought stress may be due to chlorophyll degradation and restricted $CO_2$ availability, which has been previously observed in different ornamental plant such as digitalis, callistemon, and marigold [51–53]. The reduction in *E* and *gs* is mainly attributed to decreased photosynthetic activity by stomatal closure under water stress conditions [10]. Foliar Ci application significantly enhanced *A* due to improved $CO_2$ diffusion through osmotic adjustment that also contributed to higher *E* and *Ci* under water deficit conditions [54]. In the present study, Ci also improved *gs* by enhancing chlorophyll, activities of enzymes of CAT, POX and SOD and RWC within mesophyll cells. In a previous study of Ci application on sage (*Salvia officinalis* L.), Vosoughi et al. [18] recorded higher photosynthesis in response to Ci application that contributed to increased tolerance to water stress conditions.

Water deficit conditions cause oxidative damage to plant cells due to excessive production of reactive oxygen species [55–64]. When compared to the no Ci (control), foliar Ci application significantly increased enzymatic activity (CAT, GPX, SOD) that reduces lipid peroxidation by detoxifying $H_2O_2$ and $O_2$ scavengers due to presence of abundant hydroxyl and amino groups in its structure [65]. The higher CAT, GPX, and SOD activity eliminates highly toxic effects of water stress by converting ROS species into molecular $O_2$ and $H_2O_2$. A previous report also described increased antioxidative activity in chrysanthemum [21], Greek oregano [66], and sunflower [10] in response to Ci application.

Maintenance of stomatal conductance through stomatal adjustment is considered a potential indicator of drought tolerance in plants [10]. In the present research work, water stress reduced stomatal size and density (Figure 4) at the adaxial side of the leaf that leads to loss in turgor and decreased photosynthetic activity. Stomatal adjustments are used to determine the degree of water stress as decreased stomatal size leads to reduced sites of water transpiration [67]. Foliar application of Ci significantly improved stomatal size and density showing that Ci availability improved osmotic adjustments in leaves of calendula under water stress conditions. Previously, Iriti et al. [68] observed regulation in stomatal aperture through Ci application on bean plants by regulating ABA production. According to Doares et al. [69], Ci effects the pathway of jasmonic acid production that regulates function of ABA to control stomatal functioning under drought stress.

## 5. Conclusions

The present study reported the significant impact of Ci on the physico-biochemical attributes of calendula. Optimized Ci dose (7.5 mg $L^{-1}$) considerably improved the leaf water status, membrane stability index, pigment contents, gas exchange, and stomatal size which are important parameters for water stress tolerance. Ci foliar application significantly mitigated negative effects of water stress in calendula. Hence, it may be an excellent source for water stress tolerance in floricultural crops to facilitate the floriculture business.

**Supplementary Materials:** The following supporting information can be downloaded at: https://www.mdpi.com/article/10.3390/agronomy12020474/s1, Table S1. The maximum and minimum air temperature, during the growth period of experimentation; Table S2: Number of leaves (NOL), leaf area (LA), shoot length (SL), root length (RL), shoot fresh weight (SFW), root fresh weight (RFW), shoot dry weight (SDW), and root dry weight (RDW) of *Calendula officinalis* applied with foliar application of chitosan (0, 2.5, 5, 7.5, 10 mg $L^{-1}$); Table S3: The chlorophyll (*a, b*) and carotenoid (Car) of *Calendula officinalis* applied with foliar application of chitosan (0, 2.5, 5, 7.5, 10 mg $L^{-1}$) under drought stress (60% FC).

**Author Contributions:** Conceptualization, G.A. and H.N.F.; experimentation K.R. and S.U.; write-up, F.M.W., T.J., M.A.S. and Y.S.; data analysis, M.A. and T.J.; review and editing, T.J., N.R.A., E.S.D. and M.S.C. All authors have read and agreed to the published version of the manuscript.

**Funding:** The current work was funded by Taif University Researchers Supporting Project number number (TURSP-2020/85), Taif University, Taif, Saudi Arabia.

**Institutional Review Board Statement:** Not applicable.

**Informed Consent Statement:** Not applicable.

**Data Availability Statement:** Not applicable.

**Acknowledgments:** The authors extend their appreciation to the central lab system and plant propagation and physiology lab, MNS University of Agriculture Multan, for data collection and analysis. The authors extend their sincere appreciation to Taif University for funding the current work by the Taif University Researchers Supporting Project number (TURSP-2020/85), Taif University, Taif, Saudi Arabia.

**Conflicts of Interest:** The authors declare that they have no competing interests.

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
