# Peer review of "Chitosan-Induced Physiological and Biochemical Regulations Confer Drought Tolerance in Pot Marigold (Calendula officinalis L.)"

_agronomy, doi:10.3390/agronomy12020474_

Round 1

Reviewer 1 Report

Some comments are written in memo in PDF. 

The effect of Ci on drought stress was well written.

However, the experiment methods should be clear in some ways (see memo in PDF).

One major comment is below.

Because this plant is kind of ornamental, it would be good if the impact on the flower by the drought and Ci application was investigated. 

Author Response

MNS UNIVERSITY OF AGRICULTURE, MULTAN

               Department of Horticulture

Editor-in-Chief

Agronomy

Subject: Response to Review 1

Dear Editor,

Many thanks for the consideration of manuscript “Chitosan-induced physiological and biochemical regulations confer drought tolerance in pot marigold (Calendula officinalis) (agronomy-1564094)” for publication in this Journal. The authors are also grateful to reviewers for their valuable comments and suggestions. The manuscript has been revised accordingly, and the suggested corrections/additions are listed below:

Major points

  1. Because this plant is kind of ornamental, it would be good if the impact on the flower by the drought and Ci application was investigated.
  • No doubt this is a very good suggestion to measure flower physiological and biochemical parameters affected by drought and chitosan application. However, in this study it is not possible to measure the flowers parameters at this stage. However, we appreciate the suggestion and would consider it for future studies.

Minor points

Is it true that these hormones are secondary metabolites?

  • Word has been replaced with plant growth regulators. As “It is known to increase plant growth regulators such as indole acetic acid, gibberellin, and abscisic acid that protect plants from oxidative stress and it increases crop yield”

It says the experiment was done for eight weeks in line 124. Why this table is needed?

  • The Table 1 represents the climatic data during the both experiment I and experiment II.

Is this one time application? how much?

  • Chitosan was applied two times at interval of one week. That has been explained in the text.

It needs to state the effect of the other concentration over 7.5mg/L, if there were any negative effects.

  • Thanks for comment. Only 10 mg L-1 dose showed negative results that have been discussed in the results as “Ci at 10 mg L-1 reduced shoot length (37.50%) and root length (46.25%) compared to No Ci (control). Foliar application of Ci at 7.5 mg L-1 maximum increased the shoot and root fresh weight by 60.51% and 9.49% that further decreased by 10.10% and 59.88% at 10 mg L−1 Ci respectively compared to control. The highest shoot and root dry weight by 61.41% and 22.38% was noted in Ci (7.5 mg L-1) treated plants with respect to control. Plants exhibited a marked decline in shoot and root dry weight by 12.20% and 63.46% respectively at 10 mg L-1 Ci over the control (No Ci)”

FC comes for the first time. Bring the full words in line 119 here.

  • Thanks for suggestion. Full has been added in the text.

How many times of the Ci application? 1L/pot? make it clear.

  • Sentence has been modified according to the suggestion as “Drought stress (60% FC) was applied to plants at the six-leaf stage then after a week two foliar applications of optimized chitosan level (7.5 mg L−1) were applied at one-week interval using a hand sprayer of 1 L capacity between 7 to 10 a.m”.

Best Regards,

Dr. Gulzar Akhtar

Reviewer 2 Report

This study shows that chitosan treatment improves performance of calendula plants under drought stress, including growth, photosynthesis related parameters and stress enzymes. The manuscript comprises two experiments: One to find the optimal dose of chitosan and one with chitosan foliar application under drought stress. In the second experiment, many plant physiological parameters, which are linked in the discussion, were measured.

Major points

  1. However, it did not become clear to me, why this study was conducted (other than it was not shown before in calendula). You wrote that the effects of Ci are well known (l. 72-74) and reported in various plants (l. 74-79). What is new in your study?
  2. You conclude that Ci application mitigates drought tolerance but your results clearly show that Ci application improves all measured parameters, especially growth, even more under control conditions. You should mention this in the text of the result part and discuss it.
  3. The discussion gives an unorganized impression, with many different cause-effect explanations (e.g. Ci increased enzyme activity which in turn increased chl and carotenoid concentrations (l. 375-376), gs (l. 398) and detoxify H2O2 and O2 scavengers (l.405)). I suggest to give the discussion a new structure, e.g. 1. what are the known effects of Ci application, 2. what have you shown, 3. what are the connections between the traits and with water, 4. what are the implications and achievements, 5. what are limitations, open questions for future research. Currently, I cannot see what the key result of your study is.

Minor points

  1. 36: Please give here also percentages, as the reader does not know whether those numbers (U mg-1) are high or low.
  2. 49: has been used
  3. 107: Tween-20 was added before making the dilutions? Then, the effect of optimal leaf moisturizing is also diluted.
  4. 116-117: Please indicate the Ci concentration you used here.
  5. 116/119: Explain abbreviations (FC) at first use.
  6. 118: There are 9 pots per group, but only 3 plants were harvested (l. 131). Were all measurements done on these 3 plants and how did you select the plants for harvesting?
  7. 122: Ci was applied with the start of the drought treatment, i.e. before showing symptoms. Is that realistic for practical usage? Can Ci also be used to mitigate the stress when applied when first stress symptoms appear?
  8. 124-125: What is the growth stage of the plants at harvest?
  9. 126: 2 x 2 factorial design
  10. 135: SDW and RDW
  11. 142: What do the abbreviations OD, V, W mean?
  12. 161: formula (6): LW and WL in l. 157, which one is right?
  13. 175: What is the enzyme extract?
  14. 188: were measured
  15. Table 2: I suggest to remove all fresh weight data from the results as they give no additional information and the table would look more clear.
  16. Figure 1c: Please be consistent with the letters showing significance (only here lowest number gets an a)
  17. Maybe a re-ordering of the results would give a better structure: 1st growth (3 Biomass), 2nd water (3.2, 3.5), 3rd photosynthesis (3.1, 3.3), 4th enzymes (3.4)
  18. 353: PGR, explain every abbreviation
  19. The size of all figures should be reduced and presented also side by side.
  20. Please stick to the same labels in every figure, table and the text (Fig. 4).
  21. Table 1 could be placed in the supplements. The information could be abbreviated given in the text.
  22. Table S1: Please check the numbers/unit for leaf area. The numbers are very low.
  23. Author contributions have to be specified.
  24. Acknowledgements should not be same as funding.
  25. References have to be ordered alphabetically or numbered in the text.

Author Response

MNS UNIVERSITY OF AGRICULTURE, MULTAN

               Department of Horticulture

Editor-in-Chief

Agronomy

Subject: Response to Review 2

Dear Editor,

Many thanks for the consideration of manuscript “Chitosan-induced physiological and biochemical regulations confer drought tolerance in pot marigold (Calendula officinalis) (agronomy-1564094)” for publication in this Journal. The authors are also grateful to reviewers for their valuable comments and suggestions. The manuscript has been revised accordingly, and the suggested corrections/additions are listed below:

Major points

  1. However, it did not become clear to me, why this study was conducted (other than it was not shown before in calendula). You wrote that the effects of Ci are well known (l. 72-74) and reported in various plants (l. 74-79). What is new in your study?
  • Chitosan application has been reported in different horticultural crops, but its role in mitigating water stress in calendula plants has not been studied yet. The present research work is focused on the physiological, biochemical, and anatomical alterations, vital for increasing tolerance against water stress in potted calendula plants. (Page 2, Line 82-86)
  1. You conclude that Ci application mitigates drought tolerance but your results clearly show that Ci application improves all measured parameters, especially growth, even more under control conditions. You should mention this in the text of the result part and discuss it.
  • The results of water related parameters are clearly mention in the results (Page 7, Line 53-63). The same has been discussed in the discussion section (Page 15, Line 368-391) Thanks for the nice comment. Plants were grown under normal conditions till 6 leaf stage and pots were divided in 4 groups as W for control, or normal conditions (100% FC and no chitosan), W+Ci for 100% FC and Ci, D for drought stress (60% FC and no chitosan) and D+Ci for drought stress (60% FC) and chitosan. (Page 3, Line 102-106)
  1. The discussion gives an unorganized impression, with many different cause-effect explanations (e.g. Ci increased enzyme activity which in turn increased chl and carotenoid concentrations (l. 375-376), gs (l. 398) and detoxify H2O2 and O2 scavengers (l.405)). I suggest to give the discussion a new structure, e.g. 1. what are the known effects of Ci application, 2. what have you shown, 3. what are the connections between the traits and with water, 4. what are the implications and achievements, 5. what are limitations, open questions for future research. Currently, I cannot see what the key result of your study is..
  • Authors are very thankful for such nice comment regarding discussion. However, the discussion is as per the parameters of growth, physiology and biochemical attributes. These are interrelated and discussed in cause effect manner and also in light of previous findings available on Ci.

Minor points

  1. 36: Please give here also percentages, as the reader does not know whether those numbers (U mg-1) are high or low.
  • Thanks for correction. Values has been replaced with percentage as 56.70%, 64.94%, and 32.41% (Page 1, Line 36).
  1. 49: has been used
  • Correction has been made accordingly (Page 2, Line 49)
  1. 107: Tween-20 was added before making the dilutions? Then, the effect of optimal leaf moisturizing is also diluted.
  • Thanks for correction Sentence has been corrected as Chitosan (Bio Basic Inc., Canada) was dissolved in 1% acetic acid solution and distilled water was used for making different dilutions then 0.1% Tween-20 was added as surfactant has been deleted. (Page 3, Line 107)
  1. 116-117: Please indicate the Ci concentration you used here.
  • Optimized chitosan dose (7.5 mg L-1) has been added (Page 3, Line 112)
  1. 116/119: Explain abbreviations (FC) at first use.
  • Thanks for nice comment abbreviation has been added (Page 3, Line 115)
  1. 118: There are 9 pots per group, but only 3 plants were harvested (l. 131). Were all measurements done on these 3 plants and how did you select the plants for harvesting?
  • Thanks for correction three plants were harvested from each replication (Page 4, Line 130)
  1. 122: Ci was applied with the start of the drought treatment, i.e. before showing symptoms. Is that realistic for practical usage? Can Ci also be used to mitigate the stress when applied when first stress symptoms appear?
  • Drought stress (60% FC) was applied to plants at the six-leaf stage then after a week two foliar applications of optimized chitosan level (7.5 mg L-1) were applied at one-week interval using a hand sprayer of 1 L capacity between 7 to 10 a.m. (Page 3, Line 121-123)
  1. 124-125: What is the growth stage of the plants at harvest?
  • Plants were harvested eight weeks after initial chitosan treatment at vegetative growth.
  1. 126: 2 x 2 factorial design
  • Thanks for correction, values are corrected as 2 x 2 factorial design (Page 3, Line 126)
  1. 135: SDW and RDW
  • Values are corrected accordingly (Page 4, Line 134)
  1. 142: What do the abbreviations OD, V, W mean?
  • Sentence has been modified as “High Ci levels (10 mg L-1) adversely affected growth of calendula seedlings, similarly to previous reports in Arabidopsis and cordyline (Lopez-Moya et al. 2017; El-Serafy 2020) and this may be due to modifications in auxin synthesis and cell division through modifying homeodomain transcription factor WOX5’ (Page 15, Line 347).
  1. 161: formula (6): LW and WL in l. 157, which one is right?
  • It is weight loss (WL) and has been corrected in formula at Page 4, Line160)
  1. 175: What is the enzyme extract?
  • Thanks for nice comment. Have authors have rewrite the enzyme extraction procedure as “For determination of catalase (CAT), guaiacol peroxidase (GPX) superoxide dis-mutase (SOD) activities, fresh leaf samples (0.5 g) were homogenized using mortar and pestle with phosphate buffer (pH 7.0). Then, samples were centrifuged at 1,500 rpm for 15 minutes and supernatant was separated to quantify the activities of CAT, GPX and SOD. CAT activity was assessed using the procedure of Chance and Maehly (1955) in which supernatant (0.1 ml) along with phosphate buffer (pH 7.0) and H2O2 (5.9 mM) were mixed and absorbance was read at 240 nm using the spectrometer (Hitachi-220, Japan).”
  1. 188: were measured
  • Sentence has been corrected (Page 5, Line 187)
  1. Table 2: I suggest to remove all fresh weight data from the results as they give no additional information and the table would look more clear.
  • Thanks for suggestion. All fresh weight parameters have been deleted from Table 2 results.
  1. Figure 1c: Please be consistent with the letters showing significance (only here lowest number gets an a)
  • Thanks for correction. Lettering has been corrected in Figure 1c.
  1. Maybe a re-ordering of the results would give a better structure: 1st growth (3 Biomass), 2nd water (3.2, 3.5), 3rd photosynthesis (3.1, 3.3), 4th enzymes (3.4)
  • Authors are thankful to anonymous reviewer for nice suggestion. Authors have reviewed the arrangement of different parameter in M&M, results, discussion and abstract and found it suitable for publication.
  1. 353: PGR, explain every abbreviation
  • Abbreviation has been explained
  1. The size of all figures should be reduced and presented also side by side.
  • All figures have adjusted according to suggestion.
  1. Please stick to the same labels in every figure, table and the text (Fig. 4).
  • Labels has been corrected in all figures and tables.
  1. Table 1 could be placed in the supplements. The information could be abbreviated given in the text.
  • Supplementary Table 1 has been mentioned in the text as “All growth parameters of Calendula were significantly improved in response to foliar application of different Ci levels (Supp. Table 1)”.
  1. Table S1: Please check the numbers/unit for leaf area. The numbers are very low.
  • Thanks, anonymous reviewer for nice comment. Authors have rechecked the parameter of leaf area in Supplementary Table 1. But there is still ambiguity in the parameter. Therefore, authors decided to delete the parameter from the table and the text.

  1. Author contributions have to be specified.
  • Thanks for nice suggestion. The author contributions have been specified as “Conceptualization, G.A. and H.N.F.; experimentation K.R. and S.U.; write-up, F.M.W., M.A.S. and Y.S.; data analysis, M.A. and T.J.; review and editing, E.S.D. and M.S.C.”
  1. Acknowledgements should not be same as funding.
  • Acknowledgement section has been modified as “The authors extend their appreciation to central lab system and plant propagation and physiology lab, MNS-University of Agriculture Multan for data collection and analysis. Moreover, special thanks to Taif University for funding current work by Taif University Researchers Supporting Project number (TURSP-2020/85), Taif University, Taif, Saudi Arabia”.
  1. References have to be ordered alphabetically or numbered in the text.

The revised manuscript is re-submitted to the Journal for your kind consideration.

Best Regards,

Dr. Gulzar Akhtar

Reviewer 3 Report

Dear authors,

Agronomy-1564094

The manuscript “Chitosan-Induced Physiological and Biochemical Regulations Confer Drought Tolerance in Pot Marigold (Calendula officinalis)” has been done well. However, I still have some major and minor comments for improving this manuscript.

Major comments:

  1. In line 61-62, authors should rewrite the sentence ‘decrease tissue water potential by accumulating solvents and antioxidative enzymes.’ How accumulation of antioxidative enzymes decreases tissue water potential? or authors simply want to say ‘decrease tissue water potential by accumulating solvents, and increase activities of antioxidative enzymes.’?
  2. Please explain the meaning of W and FC, as it is the first emerging time for these abbreviations.
  3. Section 2.3, it should be Chl a, and Chl b, please use italic for a and b, and also in Table 3 and Results and Discussion sections.
  4. In the section 2.4, The formula for calculation for RWC should just be water content (WC), and the calculation for RT should be the relative water content (RWC). Please check it carefully. Please change them in results and discussion sections also after checking. Turgid leaves should be leaves that after soaking in water and can’t uptake water more, so it is possible that to change turgid leaves to fresh leaves. Is it WL, but not LW in the formula for calculating ELWR?
  5. In the section 2.6, how did authors measure the protein concentration, as enzyme activities of CAT, GPX, and SOD are defined as U mg-1 of protein. Could authors explain the meaning of U?
  6. Please rewrite the Results section because I found that you have miscalculates the percentage of improvement of each parameter after foliar applied Ci compared to no-applied Ci under water stress. For example:

Line 202, line 205, line 208, line 210, and line 214-216, please recalculate the percentage of increasement about number of leaves, leaf area, highest shoot length, highest root length, Chl a, Chl b, and Car in response to 7.5 mg L-1 compared to no Ci, and Chl a, Chl b, and Car in response to 5 mg L-1 compared to no Ci as well. The difference between No Ci and 7.5 mg L-1 Ci should be divided by the value of no Ci.

Line 221, line 224, line 226, line 228, please recalculate the percentage of improvement about, the difference of value between D and D+Ci should be divided by the value of D.

Line 230, it should be 7.5 mg L-1.

Line 242, line 245, please recalculate the percentage of improvement.

Line 246, is the control (no Ci) the D condition (60% FC without Ci)? You can delete the sentence ‘Ci application enhanced Calendula plants enhanced L (27.93%), a* (37.93%), and b* (6.14).’, as you have repeatedly explained it.

Section 3.2, 3.3, 3.4, 3.5, please recalculate the percentage improvement of each parameter.

Line 269, it should be (A) water content, (B) relative water content, and please change the Fig.1A, and Fig.1B as well.

  1. Line 385-387, authors have explained the RWC, it is better to explain the MSI as well.
  2. Line 408-409, what was described in in chrysanthemum, Greek oregano, and sunflower? Authors should clearly explain it.
  3. Cite references in the text by number in square brackets, not by name and year in parentheses.

Minor comments:

  1. Is the scientific name of Calendula Calendula officinalis, but not just Calendula officinalis?
  2. Does the ‘it’ in line 57 indicates the water stress or limited water availability? If like that, I suggest authors clarify the mean of ‘it’ in line 57.
  3. Line 91, it is not necessary to mention the scientific name of calendula again, as it has been mentioned in the introduction section.
  4. In line 104, what is the mean of ‘2 foliar applications’?
  5. In line 109, it should be completely randomized design.
  6. In line 110, it should be least significance difference.
  7. Line 129, add ‘was’ before the word determined.
  8. Line 131, three plants but not ‘three plant’.
  9. Line 135, the abbreviation of shoot dry weight should be SDW.
  10. Line 141, it should be at 645, 663, and 480 nm, but not at 680 nm.
  11. In line 347-348, could authors provide some examples that high Ci levels reduced the auxin synthesis and cell division? And how high levels of Ci modify the auxin synthesis and cell division?
  12. Line 336-337, please add reference after the first sentence ‘Water stress conditions generally reduce plant biomass through decreased leaf water contents, chlorophyll concentrations, and enzymes.
  13. Line 351, should add abbreviation of PGRs after ‘plant growth regulator.
  14. Line 337 and line 358, it should be enzyme activities, but not enzymes.
  15. Line 377, delete (Shehzad et al. 2020).
  16. Line 398, change ‘enzymes (CAT, POD SOD)’ to activities of enzymes of CAT, POX, and SOD.
  17. I didn’t find the article name in the references section about Nawaz et al. 2020 in line 359, please add this article to the references section.

Author Response

MNS UNIVERSITY OF AGRICULTURE, MULTAN

               Department of Horticulture

Editor-in-Chief

Agronomy

Subject: Response to Review 3

Dear Editor,

Many thanks for the consideration of manuscript “Chitosan-induced physiological and biochemical regulations confer drought tolerance in pot marigold (Calendula officinalis) (agronomy-1564094)” for publication in this Journal. The authors are also grateful to reviewers for their valuable comments and suggestions. The manuscript has been revised accordingly, and the suggested corrections/additions are listed below:

Major points

  1. In line 61-62, authors should rewrite the sentence ‘decrease tissue water potential by accumulating solvents and antioxidative enzymes.’ How accumulation of antioxidative enzymes decreases tissue water potential? or authors simply want to say ‘decrease tissue water potential by accumulating solvents, and increase activities of antioxidative enzymes.’?
  • Thanks for nice comment, sentence has been corrected as ‘decrease tissue water potential by accumulating solvents, and increase activities of antioxidative enzymes’ (Page 2, Line 61-62)
  1. Please explain the meaning of W and FC, as it is the first emerging time for these abbreviations.
  • Thanks for the nice comment. Plants were grown under normal conditions till 6 leaf stage and pots were divided in 4 groups as W for control, or normal conditions (100% FC and no chitosan), W+Ci for 100% FC and Ci, D for drought stress (60% FC and no chitosan) and D+Ci for drought stress (60% FC) and chitosan. (Page 3, Line 102-106)
  1. Section 2.3, it should be Chl a, and Chl b, please use italic for a and b, and also in Table 3 and Results and Discussion sections.
  • Thanks for correction Chl a, and Chl b has been replaced by italic a and b throughout the research paper.
  1. In the section 2.4, The formula for calculation for RWC should just be water content (WC), and the calculation for RT should be the relative water content (RWC). Please check it carefully. Please change them in results and discussion sections also after checking. Turgid leaves should be leaves that after soaking in water and can’t uptake water more, so it is possible that to change turgid leaves to fresh leaves. Is it WL, but not LW in the formula for calculating ELWR?
  • Thanks for nice comment. According to the reviewer comment and cited reference (Redondo- Gomez et al. (2011) RWC has been replaced with WC throughout the manuscript. Moreover, authors have rechecked the RT and found it corrected according to the given reference (Clausen and Kozlowski (1965). The abbreviation LW has been corrected (WL).
  1. In the section 2.6, how did authors measure the protein concentration, as enzyme activities of CAT, GPX, and SOD are defined as U mg-1 of protein. Could authors explain the meaning of U?
  • CAT, GPX, and SOD were measured according to the procedures of Chance and Maehly (1955), Urbanek et al. (1991) and Van Rossun et al. (1997). The values are defined as U mg-1 of protein (Units mg-1 protein).
  1. 6. Please rewrite the Results section because I found that you have miscalculates the percentage of improvement of each parameter after foliar applied Ci compared to no-applied Ci under water stress. For example:

Line 202, line 205, line 208, line 210, and line 214-216, please recalculate the percentage of increasement about number of leaves, leaf area, highest shoot length, highest root length, Chl a, Chl b, and Car in response to 7.5 mg L-1 compared to no Ci, and Chl a, Chl b, and Car in response to 5 mg L-1 compared to no Ci as well. The difference between No Ci and 7.5 mg L-1 Ci should be divided by the value of no Ci.

Line 221, line 224, line 226, line 228, please recalculate the percentage of improvement about, the difference of value between D and D+Ci should be divided by the value of D.

  • The calculations are according supplementary Table 1 not shown in the main part of submitted manuscript. These has been calculated under normal conditions and are according to an already published work (Shehzad et al., 2020, Shehzad, M. A.; Nawaz, F.; Ahmad, F.; Ahmad, N.; Masood, S.; Protective effect of potassium and chitosan supply on growth, physiological processes and antioxidative machinery in sunflower (Helianthus annuus L.) under drought stress. Ecotoxicology and Environmental Safety 2020, 187, 109841.). However, if we calculate as per reviewer comment the percentage value exceeds over 100%.

Line 230, it should be 7.5 mg L-1.

  • Value has been corrected

Line 246, is the control (no Ci) the D condition (60% FC without Ci)? You can delete the sentence ‘Ci application enhanced Calendula plants enhanced L (27.93%), a* (37.93%), and b* (6.14).’, as you have repeatedly explained it.

  • Thanks for correction. Repeated sentence has been deleted.
  1. Line 385-387, authors have explained the RWC, it is better to explain the MSI as well.
  • The suggested information has been incorporated ‘Membrane stability index is wildly used to access the effects of water stress because water shortage disrupts cell membrane and internal composition (Barnabas et al. 2007)’ (Page 15-16, Line 383-388)

  1. Line 408-409, what was described in in chrysanthemum, Greek oregano, and sunflower? Authors should clearly explain it.
  • According to the suggestions statement has been cleared as ‘Previous report also described increased antioxidative activity in chrysanthemum (Elansary et al. 2020), Greek oregano (Yin et al. 2012), and sunflower (Shehzad et al. 2020) in response to chitosan application’ (Page 16, Line 410-412)
  1. Cite references in the text by number in square brackets, not by name and year in parentheses.

Minor points

  1. Is the scientific name of Calendula Calendula officinalis, but not just Calendula officinalis?
  • Thanks for correction. Complete scientific name of Calendula (Calendula officinalis ) has been added in title and introduction section.
  1. Does the ‘it’ in line 57 indicates the water stress or limited water availability? If like that, I suggest authors clarify the mean of ‘it’ in line 57.
  • Sentence has been clarified as “Water deficit conditions also produces reactive oxygen species (ROS) like hydrogen per-oxide and hydroxyl radicals that may damage proteins, lipids, and nucleic acids and ul-timately affect the photosynthetic apparatus and ATP synthesis in the plant (Shehzad et al. 2020)” (Page 2, Line 57-60)
  1. Line 91, it is not necessary to mention the scientific name of calendula again, as it has been mentioned in the introduction section
  • Thanks for correction Scientific name has been deleted. (Page 2, Line 91)
  1. In line 104, what is the mean of ‘2 foliar applications’?
  • At 6 leaf stage different chitosan (0, 2.5, 5, 7.5, 10 mg L-1) levels were applied twice at seven days interval
  1. In line 109, it should be completely randomized design.
  • Name has been corrected as completely randomized design (Page 3, Line 103)
  1. In line 110, it should be least significance difference.
  • Name has been corrected as least significance difference (Page 3, Line 104)
  1. Line 129, add ‘was’ before the word determined.
  • Sentence has been corrected according to the suggestion (Page 4, Line 127)
  1. Line 131, three plants but not ‘three plant’.
  • Sentence has been corrected according to the suggestion (Page 4, Line 129)
  1. Line 135, the abbreviation of shoot dry weight should be SDW.
  • Abbreviation has been corrected according to the suggestion (Page 4, Line 133)
  1. Line 141, it should be at 645, 663, and 480 nm, but not at 680 nm.
  • Value has been corrected as 480 (Page 4, Line 139)
  1. In line 347-348, could authors provide some examples that high Ci levels reduced the auxin synthesis and cell division? And how high levels of Ci modify the auxin synthesis and cell division?
  • Sentence has been modified as “High Ci levels (10 mg L−1) adversely affected growth of calendula seedlings, similarly to previous reports in Arabidopsis and cordyline (Lopez-Moya et al. 2017; El-Serafy 2020) and this may be due to modifications in auxin synthesis and cell division through modifying homeodomain transcription factor WOX5’ (Page 15, Line 347).
  1. Line 336-337, please add reference after the first sentence ‘Water stress conditions generally reduce plant biomass through decreased leaf water contents, chlorophyll concentrations, and enzymes.
  • Reference ‘Shehzad et al. 2020’ has been added (Page 15, Line 335)
  1. Line 351, should add abbreviation of PGRs after ‘plant growth regulator.
  • Abbreviation of PGRs has been added after ‘plant growth regulator (Page 15, Line 349)
  1. Line 337 and line 358, it should be enzyme activities, but not enzymes.
  • Name has been corrected according to the suggestion (Page 15, Line 335 and 356)
  1. Line 377, delete (Shehzad et al. 2020).
  • Reference ‘Shehzad et al. 2020’ has been deleted
  1. Line 398, change ‘enzymes (CAT, POD SOD)’ to activities of enzymes of CAT, POX, and SOD.
  • Sentence has been corrected according to the suggestions (Page 16, Line 399).
  1. I didn’t find the article name in the references section about Nawaz et al. 2020 in line 359, please add this article to the references section.
  • Thanks for correction, there was a mistake in the reference (Nawaz et al. 2020) that has been changed with ‘Usmani et al. 2020’.

The revised manuscript is re-submitted to the Journal for your kind consideration.

Best Regards,

Dr. Gulzar Akhtar

Round 2

Reviewer 2 Report

The manuscript has been improved a lot, especially the materials and methods part.

Nevertheless, some of the points have been misunderstood or should be further specified:

Major point 2:

You conclude that Ci application mitigates drought tolerance but your results clearly show that Ci application improves all measured parameters, especially growth, even more under control conditions. You should mention this in the text of the result part and discuss it.

  • The results of water related parameters are clearly mention in the results (Page 7, Line 53-63). The same has been discussed in the discussion section (Page 15, Line 368-391)
    • For experiment I, it is described that Ci application improved growth (without water stress). For experiment II, water stress without Ci was compared with control without Ci and water stress with Ci with water stress without Ci. I cannot see, where is indicated that Ci application without water stress improves growth (and other parameters) in experiment II (not in lines 253-263). This should be mentioned also for experiment II. In the conclusions, it is refered to the Ci impacts of plants under water stress. Although the study focuses on water stress, it should at least be mentioned that the same effect was also seen without water stress. I suggest: “The present study reported Ci significant impact on the physico-biochemical attributes of calendula. Optimized Ci dose considerably improved leaf water status, membrane stability index, pigments content, gas exchange, and stomatal size, which are important parameters for water stress tolerance. Ci foliar application significantly mitigated negative effects of water stress in calendula. Hence, it may be an excellent source for water stress tolerance in floricultural crops to facilitate the floriculture business.”

Minor points:

  1. 118: There are 9 pots per group, but only 3 plants were harvested (l. 131). Were all measurements done on these 3 plants and how did you select the plants for harvesting?
  • Thanks for correction three plants were harvested from each replication (Page 4, Line 130)
    • I am still not sure, what was done exactly. Is the plant material of three plants per replicate pooled, whenever plant tissue for measurements was needed? How were the three plants per replicate treated, when e.g. growth, number of leaves or photosynthesis were measured? In the tables and figures, an n=3 is given for all parameters. Please explain, maybe in the statistics section.

  1. Table 1 could be placed in the supplements. The information could be abbreviated given in the text.
  • Supplementary Table 1 has been mentioned in the text as “All growth parameters of Calendula were significantly improved in response to foliar application of different Ci levels (Supp. Table 1)”.
    • I suggested to place the table 1 with temperature data into the supplements as the information could be given in a very short sentence in the text.
  1. References have to be ordered alphabetically or numbered in the text.
    • Please, check the new line 328.
    • Additionally, the page numbering is wrong from the discussion page on.
    • 337: Supp. Table 1

Author Response

Editor-in-Chief

Agronomy

Subject: Response to Review 2 Round 2

Dear Editor,

Many thanks for the consideration of manuscript “Chitosan-induced physiological and biochemical regulations confer drought tolerance in pot marigold (Calendula officinalis) (agronomy-1564094)” for publication in this Journal. The authors are also grateful to reviewers for their valuable comments and suggestions. The manuscript has been revised accordingly, and the suggested corrections/additions are listed below:

Major points

For experiment I, it is described that Ci application improved growth (without water stress). For experiment II, water stress without Ci was compared with control without Ci and water stress with Ci with water stress without Ci. I cannot see, where is indicated that Ci application without water stress improves growth (and other parameters) in experiment II (not in lines 253-263). This should be mentioned also for experiment II. In the conclusions, it is refered to the Ci impacts of plants under water stress. Although the study focuses on water stress, it should at least be mentioned that the same effect was also seen without water stress. I suggest: “The present study reported Ci significant impact on the physico-biochemical attributes of calendula. Optimized Ci dose considerably improved leaf water status, membrane stability index, pigments content, gas exchange, and stomatal size, which are important parameters for water stress tolerance. Ci foliar application significantly mitigated negative effects of water stress in calendula. Hence, it may be an excellent source for water stress tolerance in floricultural crops to facilitate the floriculture business.”

  • According to the reviewer suggestion. Result and conclusion sections have been improved.

Minor points

I am still not sure, what was done exactly. Is the plant material of three plants per replicate pooled, whenever plant tissue for measurements was needed? How were the three plants per replicate treated, when e.g. growth, number of leaves or photosynthesis were measured? In the tables and figures, an n=3 is given for all parameters. Please explain, maybe in the statistics section.

  • Thanks for nice comment. The improved information “Eight weeks after initial treatment, plants (1 plant in each pot) were harvested for subsequent measurements. This experiment had four treatments (two factors i.e. chitosan and water regime as 2 × 2 factorial) and arranged under CRD design in three replicates” has been added (Page 3, Line 121-123).

I suggested to place the table 1 with temperature data into the supplements as the information could be given in a very short sentence in the text.

  • Table 1 has been shifted to Supplementary data and all other table numbers changed accordingly.

References have to be ordered alphabetically or numbered in the text.

Please, check the new line 328.

Additionally, the page numbering is wrong from the discussion page on.

337: Supp. Table 1

  • Thanks for correction. Reference (Page 10, Line 333) and page numbering have been corrected.
  • According to the suggestion Table 1 in the text, replaced with Supp. Table S2, S3. (Page 10, Line 342).

The revised manuscript is re-submitted to the Journal for your kind consideration.

Best Regards,

Dr. Gulzar Akhtar

Reviewer 3 Report

Dear Authors and Editor

The manuscript agronomy-1564094 has be revised well, I agree to accept is as current form.

Author Response

Esteemed Reviewer,

Thank you for your appreciated comments/suggestions for the improvement of current manuscript.

Bests,

Talha Javed